# Accelerated Skin Wound Healing Using Flexible Photovoltaic-Bioelectrode Electrical Stimulation

**DOI:** 10.3390/mi13040561

**Published:** 2022-03-31

**Authors:** Chao Han, Junfei Huang, Aodi Zhangji, Xufeng Tong, Kaige Yu, Kai Chen, Xinlan Liu, Yang Yang, Yuxin Chen, Waqar Ali Memon, Kamran Amin, Wanlei Gao, Zexing Deng, Kun Zhou, Yuheng Wang, Xiangdong Qi

**Affiliations:** 1Department of Plastic Surgery, Zhujiang Hospital, Southern Medical University, Guangzhou 510515, China; hanchao1199@hotmail.com; 2The Second Affiliated Hospital, Zhejiang University School of Medicine, Hangzhou 310009, China; 3Department of Plastic and Aesthetic Surgery, Nanfang Hospital of Southern Medical University, Guangzhou 510515, China; doctorjf@126.com; 4Frontier Institute of Science and Technology, Xi’an Jiaotong University, Xi’an 710054, China; 17791384600@163.com; 5The Faculty of Electrical Engineering and Computer Science, Ningbo University, Ningbo 315211, China; txf7255@163.com (X.T.); yukaige1999@163.com (K.Y.); chenkai2018comeon@163.com (K.C.); gaowanlei@nbu.edu.cn (W.G.); 6Medical College, Ningbo University, Ningbo 315000, China; liuxinlan0824@gmail.com; 7The Affiliated Hospital of Ningbo University School of Medicine, Ningbo 315020, China; 8Department of Plastic and Reconstructive Surgery, Zhongshan Hospital of Fudan University, Shanghai 200032, China; 20111210102@fudan.edu.cn; 9The First Affiliated Hospital of Zhejiang University School of Medicine, Hangzhou 310009, China; 10CAS Key Laboratory of Nanosystem and Hierarchical Fabrication, CAS Center for Excellence in Nanoscience, National Center for Nanoscience and Technology, Beijing 100190, China; waqaralimemon2019@nanoctr.cn (W.A.M.); mkamin92@gmail.com (K.A.); 11College of Materials Science and Engineering, Xi’an University of Science and Technology, Xi’an 710054, China; biomaterial@xust.edu.cn; 12Macromolecular Science and Engineering Center, Department of Materials Science and Engineering, University of Michigan, Ann Arbor, MI 48109, USA; 13School of Science and Engineering, The Chinese University of Hong Kong, Shenzhen 518172, China; zhoukun@cuhk.edu.cn; 14State Key Laboratory of Electrical Insulation and Power Equipment, Xi’an Jiaotong University, Xi’an 710049, China

**Keywords:** flexible photovoltaic-bioelectrode, MEMS, electrical stimulation, wound healing

## Abstract

Owing to the complex and long-term treatment of foot wounds due to diabetes and the limited mobility of patients, advanced clinical surgery often uses wearable flexible devices for auxiliary treatment. Therefore, there is an urgent need for self-powered biomedical devices to reduce the extra weight. We have prepared an electrically stimulated MEMS (Micro Electromechanical System) electrode integrated with wearable OPV (Organic photovoltaic). The wearable OPV is constructed of a bio-affinity PET-ITO substrate and a hundred-nanometer organic layer. Under sunlight and near-infrared light irradiation, a voltage and current are supplied to the MEMS electrode to generate an exogenous lateral electric field directed to the center of the wound. The results of in vitro cell experiments and diabetic skin-relieving biological experiments showed the proliferation of skin fibroblasts and the expression of transforming growth factors increased, and the skin wounds of diabetic mouse healed faster. Our research provides new insights for the clinical treatment of diabetes.

## 1. Introduction

Non-healing skin wounds such as diabetic foot ulcers, vein-related ulcers, and non-healing surgical wounds affect more than 40 million people in China and require long-term health care expenditures and, in severe cases, hospitalization of patients. The total cost of treating diabetes alone in China exceeds 600 billion RMB each year [1,2,3,4]. Acute and chronic trauma can cause severe physical pain to patients and impose a huge socioeconomic burden. Therefore, more and more attention has been paid to wound management in recent years. The main goal of skin wound management is to achieve rapid wound closure. In clinical medicine, thanks to the great progress of modern biomedicine and medical engineering technology, many effective treatment strategies, and synergistic treatment strategies have emerged, including invasive methods such as wound debridement and non-invasive techniques, conventional methods such as compression bandaging, wound dressings, hyperbaric oxygen therapy, negative pressure therapy, ultrasound, and electrical stimulation. In clinical management, passive therapeutic regimens are being replaced by regimens that can participate in the control of endogenous cell behavior [5,6,7].

Currently, advanced growth factor-mediated therapy emerges as an effective approach for regenerative skin wound healing, but still faces difficulties with nanotransport, rapid degradation, and loss of bioactivity [8]. The idea that electrical discharge stimulates cell proliferation to promote wound healing has gradually shown great potential in the field of wound management. It mimics the natural wound healing mechanism of endogenous electric fields to promote skin growth. In recent years, studies have found that an appropriate frequency of capacitively coupled electrical stimulation can promote the proliferation of skin fibroblasts, the increase in the expression of transforming growth factor β protein, the deposition and remodeling of α smooth muscle actin and type I collagen, thereby accelerating skin wound healing [9,10,11,12]. This treatment strategy has minimal side effects and is simple to apply. Flexible electronics is a solution technology suitable for wound surfaces. The latest organic optoelectronic flexible device integration innovation has opened up a way to generate periodic electrical stimulation through natural light energy [13,14,15,16]. This self-supply makes FPB (Flexible Photovoltaic-Bioelectrode) a candidate for generating self-sustaining and biologically responsive electrical stimulation. Furthermore, the electrode design of the FPB can be made flexible with rather high output voltage and sufficient energy conversion efficiency [17]. Non-fullerene materials prosperity allows for a wide choice of materials and multiple design principles [18]. Compared with excipients such as hydrogels, our device relatively takes into account repeated applications and the potential of future wearables, and realizes self-powered energy. In fact, this device can be prepared into a large area and can cover large wounds, which has practical value for acute burns, medical facial beauty and so on. Because it comes with stickers and does not need to be connected to a wired power supply, it is beneficial to the actions of patients suffering from chronic diseases, such as gout, diabetes, etc. [4]. 

Furthermore, typical organic solar cells (OSCs) are fairly stable within the range of normal biological systems under environmental conditions such as temperature and humidity. Therefore, OSCs may present an opportunity for electrical simulation-based medical treatments. In this work, we report that skin wound recovery is significantly accelerated under the influence of a power plant generated by a wearable device. We optimized the electrode shape, and with this treatment, the wound healing time in rat dorsal skin was reduced from 12 days shorten 3 days, outperforming most other reported wound recovery strategies. The remarkable effect of self-activating electrotherapy is attributed to electric field-promoted fibroblast migration, proliferation, and transdifferentiation.

## 2. Materials and Methods

### 2.1. Small-Area Organic Solar Cells Fabrication

PTB7-Th was obtained from 1-material Chemscitech Inc. (Dorval, QC, Canada), whereas COi8DFIC and PC_61_BM were obtained from Hyper, Inc. (ShenZhen, China). All of the supplies were utilized exactly as they were supplied. The glass–ITO substrate (15 sq^−1^) used in spin coating devices was bought from South China Xiang Science & Technology Co., Ltd. (ShenZhen, China) for use in spin coating devices. The ITO glass was cleaned using a three-step process that included sonication in soap and deionized (DI) water, acetone, and lastly isopropyl alcohol for 20 minutes each. After a 10-minute treatment with ultraviolet–ozone (Ultraviolet Ozone Cleaner, Jelight Company, Irvine, CA, USA), a ZnO electron transport layer was spin-coated over the ITO substrate at 3000 rpm. ZnO sol–gel with isopropyl alcohol solvent was prepared at a concentration of 15 mg mL^−1^ and dissolved in isopropyl alcohol. The active layer solutions were produced in o-xylene with a solvent addition of 1.5 percent *v*/*v* 1,8-diiodooctane (DIO) to provide a solvent for the active layer solutions. When forming the blends, a polymer concentration of 10 mg mL^−1^ was employed while keeping the total PTB7-Th:COi8DFIC:PC_61_BM ratio constant at 1:1.2:0. 5 weight percent (*w*/*w*). At ambient temperature, the active layers were spin-coated at 2500 rpm in a N_2_ glove box with a nitrogen atmosphere. A thin layer (5 nm) of MoOx was deposited as the anode interlayer at a vacuum level of 1.0 × 10^−6^ mbar, and a 160 nm Ag layer was deposited as the top electrode The spin-coated devices had an area of 0.04 cm^2^ and were made of polymer.

### 2.2. Large-Area and Module Flexible Organic Solar Cells Fabrication

To create the bottom electrode, we used the silver-grid patterned on PET substrates (1.5 sq^−1^, T ≥ 86%) that were created by ourselves with help from factories. The pattern was conceived and processed by the collaborators. The mini-roll R2R coater (FOM Technologies, Denmark) was used. The PET silver-grid/ZnO/active layer/MoOx/Ag structure was used to fabricate the inverted solar cell devices, which were produced in an ambient environment. To increase the surface flatness and wettability of the substrates, hybrid electrodes made of PET, silver grid, and PH1000 were developed and tested. PET/silver-grid substrates were coated with highly conductive poly(3,4-ethylenedioxythiophene) polystyrene sulfonate (PEDOT:PSS), which was diluted with isopropyl alcohol at a ratio of 1:3 *v*/*v* (Clevios PH1000, Heraeus, Hanau, Germany) and then SD(Slot-Die Coating) coated with an injection speed of 10 mL h^−1^ and a roller speed of 35 m h^−1^ The hybrid electrodes were roasted at 120 degrees Celsius for 15 minutes in room temperature air. In the next step, ZnO was coated to serve as the electron transport layer (ETL) using an SD coating ZnO sol–gel with isopropyl alcohol solvent (15 mg mL^−1^), and the solution flowed through the roller at the rate of 4 milliliters per hour. o-Xylene was used to create the active layer solutions, which had polymer concentrations of 10 mg mL^−1^ and an additive solvent of 1.5 percent *v*/*v* 1, 8-diiodooctane (DIO) was added as the additive solvent. The weight-to-weight ratio of PTB7-Th:COi8DFIC:PC_61_BM was maintained at 1:1.2:0.5 weight-to-weight. In ambient circumstances, after heating the roller to 80 degrees Celsius, the active layers were slot-die coated at an injection rate of 8.0 milliliters per hour and a roller speed of 65 millimeters per hour. After that, the samples were moved into a vacuum evaporation room, where the top electrode was deposited. Using different-shaped templates, a thin coating of MoOx (ten nanometers) and Ag (one hundred and sixty nanometers) was thermally evaporated onto the samples at a vacuum level of 1.0 × 10^−6^ mbar. Single cells had an active area of 0.04 cm^2^ and 1.25 cm^2^, whereas modules had active areas of 2.5 and 5.0 cm^2^. The modules are connected in series using conductive silver paste. PET–ITO substrates were obtained from FOM Technologies in Denmark for use in flexible devices based on PET–ITO substrates. The PET–ITO substrates have a square resistance of 30 sq^−1^, while the thickness of ITO is around 80 nm. The remaining processes are the same as previously, with the except for coating PH1000 and annealing for the bottom electrode.

### 2.3. Wound Electrode Fabrication

Flexible MEMS electrodes are prepared in several parts. The first step is to deposit UV photoresist on top of the glass substrate with a doctor blade and pattern the UV photoresist film by photolithography. Take the hexagon as an example, the designed hexagonal honeycomb structure can transmit light in the maximum range. The metal grid covers only about 3% of the substrate, resulting in fairly low optical losses. Such a low light loss results in a high light transmittance (80–90%) of this flexible electrode, which is only about 3% lower than that of the PET substrate, which helps to improve the translucent cell transmittance. The diagonal distance of the hexagon is 200 µm. We also designed cross-shaped and serpentine shapes with different degrees of coverage, to obtain a fractal shape suitable for the wound surface. The nickel master is then fabricated by using the already patterned UV photoresist film as a mask. The second step is the patterning of the PET substrate: UV glue is knife-coated on top of a 120 µm thick poly(ethylene terephthalate) PET substrate, and then a nickel master is imprinted on the PET by imprinting a nickel master on top of the UV glue A mask-like pattern was formed on the grid, and the grid reserved thickness was 3 µm. The third step was to fill the silver nano ink into the grooves by using a scraper, and then sintered at 150 °C for 15 minutes, and placed on top of the conductive silver grid. The copper grid was electroplated for about 5 minutes, and the electroplating current was 2A. By electroplating to create a dense microstructure of the surface Cu layer, further oxidation of the electrodes may be passivated. Finally, the film is polished using an aqueous solution of silica particles to smooth the surface.

## 3. Results

The self-powered electrical stimulation device consists of two parts: a natural photoelectric energy conversion part (i.e. OSCs) and a flexible dressing electrode. As shown in Figure 1, OSCs were fabricated by slot-die coating of organic semiconductor materials on a polyethylene terephthalate (PET) substrate. Due to the good flexibility of PET loaded with silver grids (Young’s modulus of 0.45 MPa), the silver grid composite substrate devices (PET-OSCs and PET-electrodes) show lower flexural modes compared to pure PET films amount and shows good adaptation to soft skin surfaces. This organic solar cell is designed to work well under natural daylight conditions, and its voltage. Its biotoxicity was assessed by measuring the cell viability of 3T3 fibroblasts cultured on the surface of PET substrate devices using live-dead staining. Results up to 72 h showed that the device surface had a negligible effect on cell viability (Appendix A), implying that they could be safely applied to wounds.

Compared with devices fabricated by spin-coating processes on small areas, devices fabricated on flexible large-area substrates by thermal slot-die coating commonly used in the industry can achieve the high efficiency of small-area rigid devices used in laboratories. Our previous work has demonstrated that synergistic optimization of organic layer morphology, flexible substrate properties, and processing conditions can fabricate the desired devices. Serial modules are prepared to ensure sufficient electrical stimulation voltage. Single-junction large-area cells are connected in series through conductive silver paste to form a module.

In this work, the PTB7-Th:COi8DFIC:PC_61_BM-based ternary system was employed due to its good solubility in non-halogenated solvents and its high efficiency. Compared with the devices fabricated at room temperature, the SD-coated film showed a larger redshift, and the absorption peak at 850 nm (the main absorption peak of the COi8DFIC film) broadened, indicating the formation of J-type π-π stacking, which is consistent with the reported The results are consistent.Large-area single-junction flexible OSCs (1.25 cm^2^) maintained 85% PCE of small-area (0.04 cm^2^) rigid devices (12.16%). In addition, a 5 cm^2^ large-area module was fabricated, which achieved an excellent performance of more than 7% The efficiency is also maintained at 85% PCE for single-junction large-area flexible devices. The absorption spectra of organic thin films formed by ambient and thermal SD coating methods at different temperatures are shown in Figure 2a. The flexible OSC showed good flexibility and stability under a bending radius of 10 mm, which could be maintained at more than 95% to cope with the movement of the organism itself. The specific performance is shown in Table 1.

Initial testing consisted of circular wounds (1 cm^2^ in size) made on the back of anesthetized rats. Photovoltaic stimulation device: Contact electrodes (front) and self-functional modules (back) are wrapped around the body, with the dressing electrodes facing the wound. Based on the effect of endogenous electric fields on wound site recovery, self-activating electrotherapy is hypothesized to accelerate wound recovery. As shown schematically, in the injured area, disruption of the transepithelial potential (TEP) induces an endogenous electric field that has been previously identified as a signal for epithelial cells to initiate directed migration to the skin wound bed. This potential is maintained until the skin regeneration process is complete [19,20]. In our design, dressing electrodes are placed on top of the wound to generate an electric field that can penetrate the dermis and enhance the endogenous electric field to promote wound healing. The flexible organic solar cells pasted on the surface of biological skin as shown in the Figure 3. We did preliminary experiments on cells and verified that the effect is feasible, shown as Appendix A [21,22].

We designed a separate animal experiment using surface electrodes [23,24]. This modality is now widely accepted ES therapy. Compared with insertion electrodes, surface electrodes are noninvasive and more convenient to use; however, suitable electrode contact shapes have not been explored. We designed three surface electrode shapes, and different shapes have different contact areas, so the compensation potentials acting on the subcutaneous surface are different. As shown in Figure 4. A module consisting of 4 single-junction cells is used to provide high penetration voltage, which is normalized during simulation. The connection method is shown in Appendix A. In the literature this voltage is large enough to bring about a biological response. The side length of the cross electrode is 0.18 mm, the side length of the hexagonal electrode is 0.1 mm, and the side length of the serpentine electrode is 0.2 mm. The actual side width of a hexagon is a multiple of the side length. Here the conductivity of the PET is set to be 10^−20^ S/m, electrode sliver mental 58^6^ S/m, hydrogel 100 S/m and human skin 10^−4^ S/m.

In order to compare the therapeutic effects of different treatment plans, photographs of each group were taken at different time points, and quantitative analysis was performed with ImageJ software to evaluate the healing effect after transplantation. As wound healing progresses, the resistance of the wound surface (primarily the epidermis) increases significantly. Inserted electrodes will have the ability to apply voltage directly to the tissue without generating a current through the epidermis. The normalized simulation results show that the intersecting electrodes are the densest, the hexagons are the sparsest, and the serpentines are more balanced. Application of all three electrodes resulted in accelerated wound closure (Figure 5a). Compared with the results of all other groups, the wounds of the rats treated with the serpentine electrodes healed faster than the control wounds. Therefore, we conclude that serpentine electrode is the most efficient of those electrical parameters evaluated.

Scarring is an inevitable consequence of full-thickness skin injury. This process is characterized by reorganization of the extracellular matrix (ECM) and loss of normal skin appendages such as hair follicles. H&E staining (Hematoxylin & Eosin staining) showed that the healed wounds treated by serpentine electrode stimulation were smaller than the control group. This also indicated that serpentine electrode had better wound healing outcomes. In addition, wounds treated with serpentine electrode stimulation expressed higher levels of growth factor (PCR) compared to controls, this factor is thought to be a key activator of angiogenesis during wound healing.

As shown, the wounds in the experimental group produced more epidermal tissue compared to the control group. H&E staining showed that the experimental group developed good epidermal tissue compared to what was observed in the control group 10 days after surgery. At this time, the epithelium in the serpentine electrode-treated group was almost completely closed. This suggests that wound epithelization may be accelerated under conditions where cells present an appropriate microenvironment. After healing, the visible scars in the rats treated with the serpentine-shaped electrode dressing were significantly smaller than those in the control group. Use the cross and hexagon to make the scar smaller. These results suggest that scarring can be inhibited by electrode-compensated electric fields. 

We collected newly formed skin for histological study by hematoxylin and eosin (H&E) staining. H&E staining results showed that the width of the dermal layer after healing was greater in the rats treated with ES compared to the control group. Compared with the control group, inflammation was significantly reduced after electrode treatment. Part of the skin appendages were born, the epithelium was partially completed, and the wound was closed, with hair follicles close to healthy tissue. The same trend was also seen in the pathological index analysis, where the serpentine electrode physiological index was significantly better than the other groups). Since the regeneration of collagen fibers is another important indicator of skin regeneration, new tissues from all groups were collected for Masson staining. As shown, all groups showed increased collagen deposition. The serpentine electrodes exhibited denser and more organized collagen deposition in which mature collagen fibers formed, whereas control and untreated collagen fibers remained partially stunted. Inflammation levels also greatly affected the regeneration of collagen fibers. The results of PCR indicated that gene expression of type I collagen I (Col I) and platelet-derived growth factor (PDGF) was significantly enhanced in groups treated with electrical therapy. Collagen was known to play an important role in wound healing. During the proliferation phase of wound healing, abundant Col I secreted by fibroblast enhance the skin structure and its integrity. In addition, the well-organized collagen fibrils network serves a stent for cell migration, re-epithelialization and angiogenesis. It is found that electrical stimulation can promote the proliferation of fibroblasts and increase the content of collagen, thus promoting the wound healing process [25]. Additionally, PDGF is a dimeric polypeptide regulating multiple cell function such as cell proliferation and differentiation. It is documented that PDGF involved in re-epithelialization, extracellular matrix deposition and angiogenesis via activation of extracellular signal-regulated kinases (ERK) pathway in the healing process of cutaneous tissues [26]. Our result showed that both enhanced Col I and PDGF gene expression levels were detected in mice treated with electrotherapy via hexagonal and serpentine electrodes, indicating the profound therapeutic effect generated by specific flexible electrodes.

Similarities exist between the normal healing process of acute wounds and the electrotherapy healing process of chronic wounds. Both involve cell migration and proliferation during healing, which can be facilitated by electrical stimulation. Electrical stimulation has been reported to promote wound healing by accelerating keratinocyte and macrophage migration, improving angiogenesis, stimulating fibroblasts and collagen regeneration. Appendix A shows the simulation results of the skin microstructure against the electrode-skin conductivity in different states. The electric field and potential distribution within the skin were compared when the three electrodes were applied to the skin. Appendix A–c show the potential distribution in the skin after the voltage is applied to it, respectively. It can be seen that the flexible metal electrodes can generally be closely attached to the skin, and the potential distribution is uniform down the vertical direction of the skin. It can be seen that under the same conditions, the serpentine electrode has low stress, high load-bearing capacity and strong tensile strength, which means that it is extremely suitable for the application environment that fits the skin. In contrast, hexagonal electrodes can only be connected to the protruding part of the skin, and the potential distribution in the recessed area is smaller. As shown in the electric field distribution shown in the figure, it can also be seen that the serpentine electrodes are more uniformly distributed. At the same time, since the direction of the electric field is the same as that of the current, and its magnitude is proportional, the use of serpentine flexible electrodes can achieve a more uniform effect regardless of whether the current is transmitted inward or outward. In terms of safety, the crossed electrodes make too close contact with the skin. As a result, under the high impedance formed by the local air, a large amount of heat is generated to burn the skin, as shown by the bright spots in the figure. We finally added immunohistochemistry (Figure 6) to further validate the rationale of the strategy [27,28].

## 4. Conclusions

In this study, we have identified fractal MEMS electrodes as an effective parameter for ES treatment of diabetic wounds, and we conduct a preliminary investigation of the underlying molecular mechanisms. Furthermore, we fabricated a flexible wearable electrical stimulation device of OSC-MEMS; this approach promotes diabetic wound healing by accelerating angiogenesis, enhancing epithelialization, and inhibiting scarring. This dressing may be used in future clinical applications, especially for the treatment of diabetic wounds. In future research, the dressing could contain sensors that help quantitatively assess wound condition. Our animal models provide “value information”.

## Figures and Tables

**Figure 1 micromachines-13-00561-f001:**
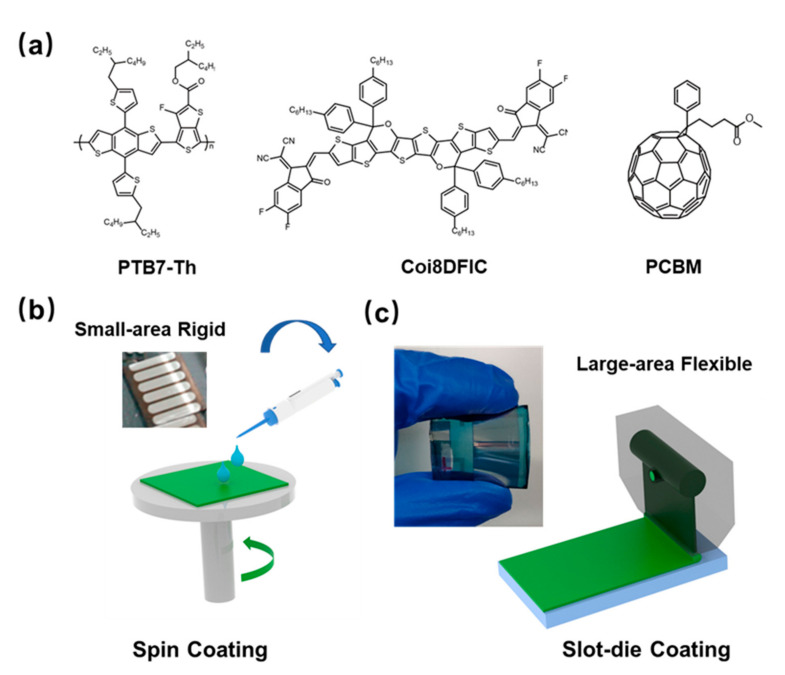
(**a**) Chemical structures of PTB7-Th, PC_61_BM and COi8DFIC. (**b**) Schematic of spin-coated and small-area rigid devices. (**c**) slot-die process and large area flexible devices and large area flexible devices.

**Figure 2 micromachines-13-00561-f002:**
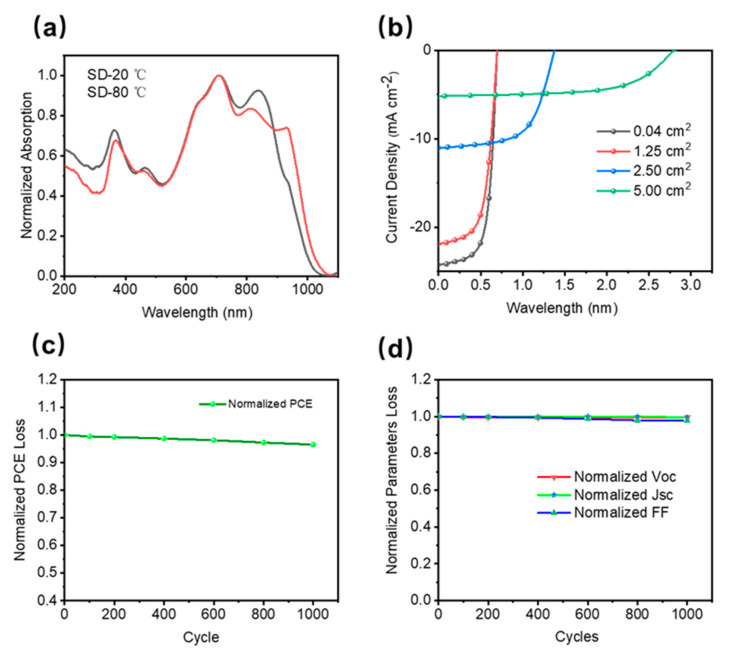
Large-area flexible solar cell structure: PET(Ag-grid)/ZnO/, PTB7-Th:COi8DFIC:PC_71_BM/MoO_3_/Ag (**a**) Normalized Absorption of different slot-die temperature. (**b**) 0.04~5 cm^2^ large-area rigid/flexible solar cells/modules with different areas The JV features. (**c**) Bend test normalized PCE characteristics. (**d**) Bend test normalized electrical loss characteristics. (Average performance is statistically derived from testing the same batch of devices. The margin of error is within 5%).

**Figure 3 micromachines-13-00561-f003:**
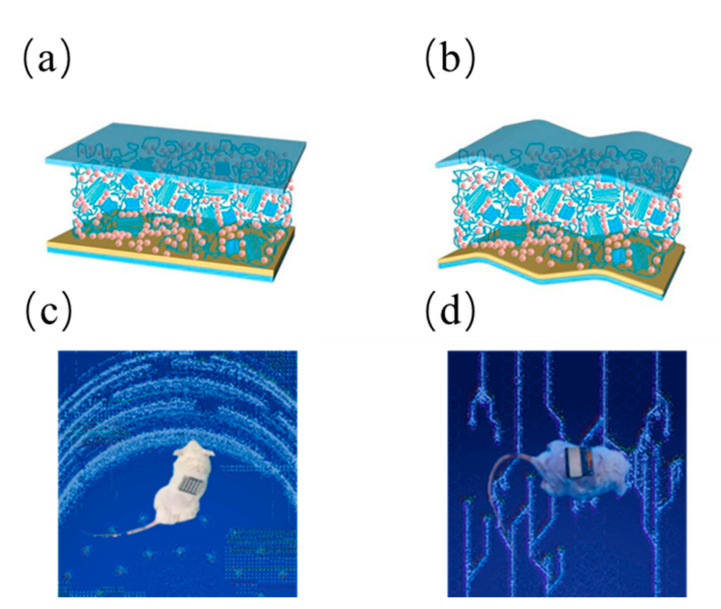
(**a**) Schematic diagram of rigid device morphology (**b**) Schematic diagram of flexible device morphology (**c**) 0.04 cm^2^ device on animal photo (**d**) 1.25cm^2^ device on animal photo.

**Figure 4 micromachines-13-00561-f004:**
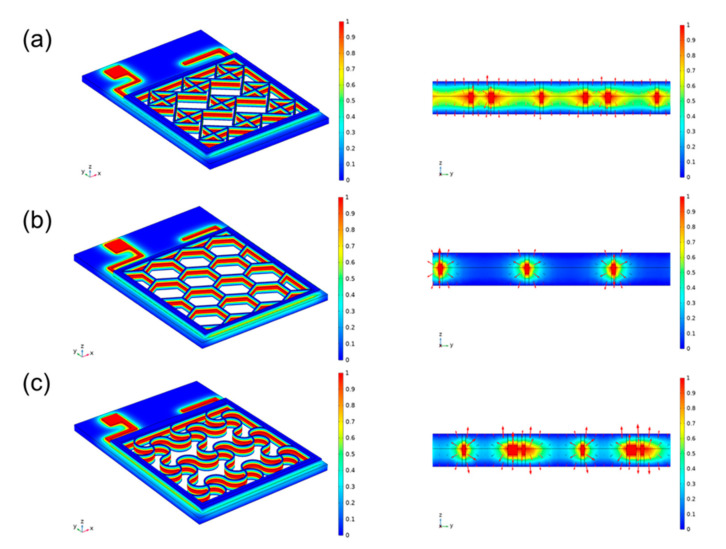
Experimental Section MEMS electrode design left (**a**) cross electrode (**b**) hexagonal electrode. (**c**) serpentine electrode Right Side view of electric field distribution of the combined PET-electrode-hydrogel-skin model.

**Figure 5 micromachines-13-00561-f005:**
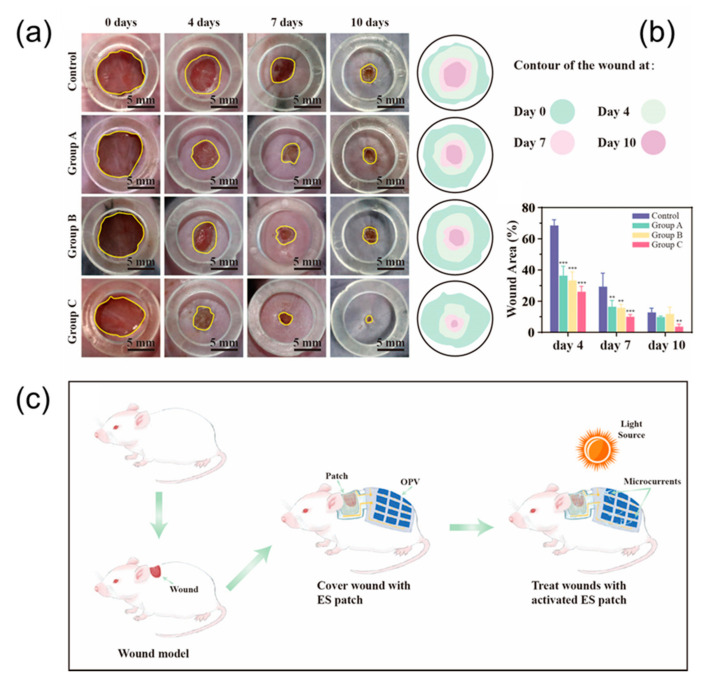
(**a**) Photographs of wound regions with different treatments and time points. (**b**) Analysis of wound closure. Data shown as mean ± SD. N = 4 per group. ** means *p* < 0.01, and *** means *p* < 0.001. (**c**) Schematic diagram of the integrated integration of MEMS electrodes and OSCs self-powered modules on animals.

**Figure 6 micromachines-13-00561-f006:**
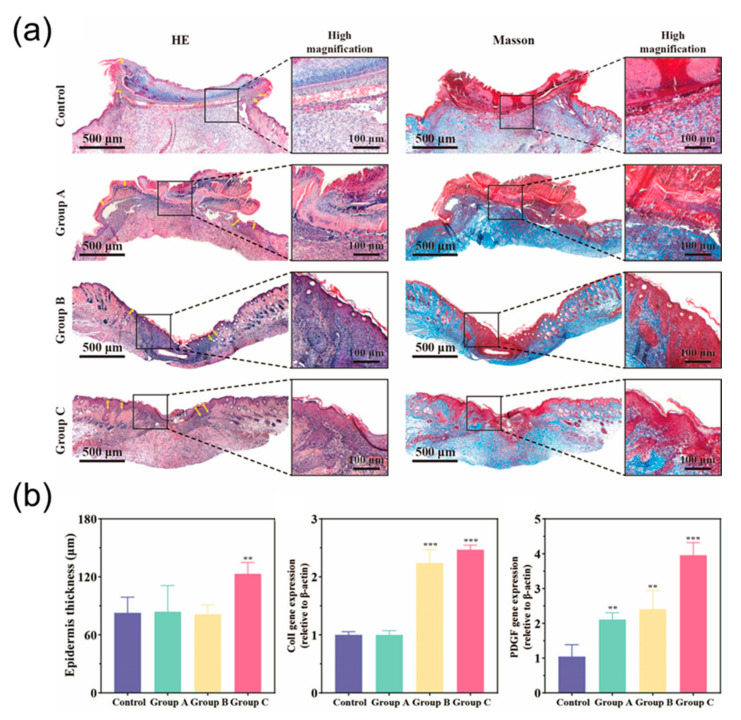
(**a**) H&E and Masson staining for different treatments and periods. The scale bar is 500 µm. (**b**) Pathology index: epidermis thickness, coll gene expression and PDGF gene expression measured by qPCR. Data shown as mean ± SD. n = 4 per group. ** means *p* < 0.01, and *** means *p* < 0.001.

**Table 1 micromachines-13-00561-t001:** Involved module parameters and photovoltaic performance (Average performance is statistically derived from testing the same batch of devices. The margin of error is within 5%).

Number of Sub-Cells	Area [cm^2^]	Voc [V]	Jsc [mA cm^−2^]	FF [%]	PCE [%]
1	0.04	0.69	24.2	65	10.85
1	1.25	0.69	21.8	62	9.32
2	2.50	1.36	11.0	58	8.68
4	5.00	2.77	5.1	56	7.91

## Data Availability

The data that support the findings of this study are available from the corresponding author upon reasonable request.

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
