# Peer review of "Accelerated Skin Wound Healing Using Flexible Photovoltaic-Bioelectrode Electrical Stimulation"

_micromachines, 2022, doi:10.3390/mi13040561_

Round 1

Reviewer 1 Report

In this manuscript, the authors developed a flexible bioelectronic based on organic photovoltaic polymers for light-mediating electrical stimulation to promote wound healing. The design and findings in this work is interesting. however, major improvements shall be made for the current manuscript. Detailed comments are listed below:

  1. In Introduction, the argument that conventional approaches for promoting wound healing including electrical stimulation are “primarily passive treatments with little involvement in the control of endogenous cellular behavior” is arbitrary.

  1. In Introduction, can the authors explain the advantage of light-mediating electrical stimulation over other approaches, for example, biologically responsive electrical stimulation, in promoting wound healing?

  1. In Experimental section, the authors shall provide detailed procedures for the experiments of cell culture, cytocompatibility assay, animal surgery, and histological staining.

  1. The authors shall provide the wavelength, power density, and exposure time of the light used during evaluating the photovoltaic performance.

  1. Please provide the error bars for the data shown in Fig. 2b, 2c, and 2d, and Table 1.

  1. The authors shall compare the mechanical properties among the bioelectronics with different electrode layout designs.

  1. In Introduction, the authors mentioned many times “non-healing wound treatment”. However, the animal model used is not for non-healing wound.

  1. Can the authors discuss the possible signaling pathways that affect the expressions of collagen (the authors should also indicate that which type of collagen investigated in this study) and PDGF by the electrical stimulation exerted by the bioelectronic?

  1. The authors should discuss the crosstalk between the gene expression (Col, PDGF) with the tissue regeneration during wound healing.

  1. The authors shall carefully revise throughout manuscript. There are too many errors in grammar making the sentences difficult to understand.

Author Response

Thank you very much for your review,

Reviewer 2 Report

The results from the in vitro and in vivo animal experiments should be added to the main article not as a supplementary material (except if the results have been published already).

FIgure legends should talk to the figures described. For example, Figure 5 has no non significant statistics (NS) shown but the legend says otherwise 

IN Figure 6: there is no NS on the figure but it appears on the figure legend. 

The authors mentioned that the the murine fibroblast cell line NIH-3T3, was cultured in 0.2–0.4 mM BSA-conjugated palmitate to stimulate a high fat environment, please give relevant citations to this so as to make the research reproducible. 

Author Response

Thank you very much for your review 

Reviewer 3 Report

In this draft paper, the authors proposed a flexible device that can accelerate the skin wound healing by photovoltaic electrical stimulation. The proposed method has great potential against various applications, and the animal test have provided good support on the performance of the proposed device. Having said that, extensive editing and revision is required to improve the readability of this manuscript. I think the manuscript could be suitable for publication in Micromachines after addressing several concerns bellow.

  1. Fig 5, any data at day 0? Also, for control group, do you have wound model with light source, but without ES patch attached?
  2. What current can be generated with three design you provided, under the light source? Is there any characterization regarding the light intensity? The prospective electric field/voltage that will be generated? What condition is used in the model simulation? How can we compare them to justify the assumption?

Minors:

  1. A deep-through proof reading is required to fix the spelling, punctuation, and grammar errors. For example (here I just list several but not all),
    1. “PC61BM” vs “PC61BM”, “pathology” vs. “physiology”;
    2. missing punctuation at line 197, line 200;
    3. “um” should be “µm”;
    4. Figure 3a labeled as 3c, Fig S6, “a)” is without a sub-caption labeling at the figures;
    5. “cm 2”, “2” should be superscribe
    6. Grammar error line 142, 197,276, etc.
    7. No proper sub caption for most supplementary figures.

Please revise accordingly.

  1. Please be consistent with the naming of your structure. “serpentine” vs. “snake”, please choose one and replace the other term.
  2. Abbreviation in the manuscript: please have the full name when the first time you have the abbreviation, for example, “SD” or “H&E”, and terms in Table 1 (Voc, Jsc, FF, PCE). It helps reader to better understand the presented work.
  3. Figure captions should contain detailed information on what contents are presented, so that the figure itself could be more self-explanatory. In most figure, by just looking at the figure caption, it is quite hard to comprehend the data.
  4. 3, here the generated electric fields are normalized and displayed. However, in this case, the cross comparison between different designs can not be done with this normalized scale. I would suggest displaying the original intensity as what is done in supplementary materials.
  5. Line 292, “PCR” should be “qPCR”?
  6. Line 377, please use phrase as title, not sentence
  7. If the initial testing wound was designed to be 1cm2, how a 0.04cm2 patch design can be attached and photos as in Fig 4c?

Author Response

Thank you very much for your review

Round 2

Reviewer 1 Report

The authors have carefully revised the manuscript and responded all the comments appropriately. 

This manuscript is a resubmission of an earlier submission. The following is a list of the peer review reports and author responses from that submission.